# Determining behavioral intention and its predictors towards cervical cancer screening among women in Gomma district, Jimma, Ethiopia: Application of the theory of planned behavior

Wadu Wollancho[1], Demuma Amdissa[2], Shemsedin Bamboro[2], Yitbarek Wasihun[2], Kasahun Girma Tareke[2]*, Abraham Tamirat Gizaw[2]

1 School of Nursing and Midwifery, Institute of Health, Jimma University, Jimma, Ethiopia, 2 Department of Health, Behavior, and Society, Institute of Health, Jimma University, Jimma, Ethiopia

* kasahungirmadera@gmail.com

**Data Availability Statement:** All relevant data are within the manuscript.

## Abstract

### Background

Cervical Cancer is the leading cause of cancer-related deaths among Ethiopian women. Despite many interventions were conducted, there is low uptake of cervical cancer screening services. Also, limited evidence was available on the women's intention and its predictors towards cervical cancer screening. Therefore, this study was aimed at determining the intention and predicators of behavioral intention toward cervical cancer screening.

### Methods

A cross-sectional study was conducted in the Gomma district, Jimma, Ethiopia from August 1–30, 2019. The total sample sizes were 422 and a systematic random sampling technique was employed to select the samples. Data were collected through interviews using a structured questionnaire guide. Data were entered in epidata, and exported and analyzed using SPSS version 20.0 software. Descriptive, correlation, and multicollinearity analysis were done. Also, simple and multiple linear regression analysis were performed to identify the predictors for behavioral intention. The p-value<0.05 was used to declare a significant association.

### Result

The response rate was 382 (90.5%). The mean age of the participants was of 26.45 (SD = 4.76). Direct attitude, subjective norm, and perceived behavioral control had a mean score of 16.78 (SD = 2.87), 15.61(SD = 1.92), and 12.86 (SD = 4.85), respectively. The intention has a mean score of 14.52 (SD = 4.01). From regression analysis, direct attitude (B = 0.346, p<0.001), direct subjective norm (B = 0.288, p = 0.008), direct perceived behavioral control (B = 0.132, p = 0.002) indirect attitude (B = 0.015, p = 0.019) and the indirect perceived behavioral control (B = 0.132, p = 0.002) were statistically significant with intention.

**Funding:** The author(s) received no specific funding for this work.

**Competing interests:** The authors have declared that no competing interests exist.

**Abbreviations:** BCC, Behavior Change Communication; HEW, Health extension worker; HPV, Human Papilloma Virus; LMIC, Low and middle-income country; PCA, Principal component analysis; TPB, Theory of planned behavior; WHO, World Health Organization.

## Conclusion

From this study, it was understood that women's intention towards cervical cancer screening was low. The predictors were the direct and indirect attitude, direct and indirect subjective norm, direct and indirect perceived behavioral control. This calls a need to develop strategies and take action to improve the attitude of women and their influential peoples and increase sense of control to improve their intention to screen for cervical cancer. Moreover, health care providers should have to conduct social and behavioral change communication to improve women's health seeking behavior towards cervical cancer screening applying the concept of theory of planned behavior.

## Background

Globally, in 2018, cervical cancer was the fourth most common cancer among women with 570,000 new cases and 311,000 deaths (7.5% of all female cancer deaths) [1–3]. In America, it is one of the leading causes of cancer deaths among women with an estimated number of 83,200 newly diagnosed and 35,680 deaths per year [4]. It is also one of the leading causes of cancer deaths among women in low and middle-income countries (LMIC), where 83% of new cases and 85% of related deaths occur in these poor resource countries; affecting poor, vulnerable, and disenfranchised women at the prime of life [1]. In Ethiopia, one of the LMCIs, cervical cancer is the commonest cancer affecting reproductive organs and also the leading cause of death from cancer among women [5]. In the country, an estimated number of 6,300 new cases diagnosed annually, and about 4,884 women die from cervical cancer each year [6].

Even though the disease is a cause for the morbidity and mortality of these women, it is one of the most preventable and curable forms of cancer, as long as it is detected early and managed effectively [1]. However, the majority of cases (over 80%) in sub-Saharan Africa are detected at a late stage, predominantly due to a lack of information about cervical cancer and the scarce of prevention services. This advanced stage by itself needs to involve multiple treatment modalities including surgery, radiotherapy, chemotherapy, or mostly is a stage where treatment is likely lacking/limited, ineffective, too expensive or inaccessible for many women in low-resource countries, including Ethiopia, or it is associated with a markedly diminished chance of prognosis successes after treatment [5, 7, 8].

To overcome such challenges, the world health organization (WHO) developed a global strategy to eliminate cervical cancer among women of all countries in 2030. This will be achieved through provision of full human papillomavirus (HPV) vaccine to 90% girls by age of 15 years (i.e. primary intervention), conducting screening for 70% women 15–45 years of age, and providing treatment and care for 90% identified women (i.e. secondary intervention) [1]. Our country, Ethiopia, also developed the guideline to prevent and control cervical cancer through primary prevention, secondary prevention, and tertiary care as well as palliative care. Health facilities are responsible for the implementation and to design appropriate communication and advocacy strategies to increase the utilization of cervical cancer promotive, preventive and curative health services. Behavioral Change Communication (BCC) intervention is one activity conducted to increase awareness of cervical cancer prevention, influence social norms, and facilitates behavior change amongst selected individuals or sub-populations to prevent cervical cancer. Moreover, the health extension workers and health development armies are an essential part of the community who promote the acceptability of cervical cancer prevention

services. They play a role in advocating for and providing information about cervical cancer prevention services, identifying the eligible groups, and assisting women in making decisions to attend the health facilities for cervical cancer prevention services and engaging cervical cancer survivors [5].

Despite this case, the findings of a study conducted in different settings indicated that there was low uptake of cervical cancer screening services [9–14]. Lack of awareness, poor attitude towards cervical cancer screening, and poor perception of the severity of the disease were some of the factors for not up taking the screening service. To produce a significant decrease in incidence and mortality related to cervical cancer, by increasing the uptake of cervical cancer screening service, there is a need to address such barriers. Thus, awareness should be created, and there must be effective screening and prevention services that facilitate early detection and treatment. To solve this problem, BCC interventions were conducted in Gomma district, Jimma, Ethiopia, from 2018–2019 to create awareness and increase women's health-seeking behavior towards cervical cancer screening at an early stage. Therefore, this study was designed to determine women's intention and its predictors towards cervical cancer screening using the theory of planned behavior (TPB) framework.

## Methods

### Study design, setting and period

The cross-sectional study design was employed. The study was conducted in the Gomma district, Jimma zone, Oromia regional state, Ethiopia from August1-30, 2019. The district is bordered on the south by Seka Chekorsa, on the southwest by Gera, on the northwest by Setema, on the north by the Didessa River which separates it from the Illubabor Zone, on the northeast by Limmu Kosa, and on the east by Mana. The altitude of this district ranges from 1,380 to 1,680 meters above sea level; however, some points along the southern and western boundaries have altitudes ranging from 2229 to 2870 meters. A survey of the land in this woreda shows that 60.7% is arable or cultivable (52.7% was under annual crops), 8.1% pasture, 4.6% forest, and the remaining 20.1% are considered swampy, mountainous or otherwise unusable. Land in cultivation included the two-state coffee farms. The 2007 national census reported a total population for this district of 213,023, of whom 108,637 and 104,386 were men and women, respectively. About 12,769 or 5.99% of its population were urban dwellers. The majority of the inhabitants were Muslim; with 83.88% of the population reporting they observed this belief, while 14.68% of the population said they practiced Ethiopian Orthodox Christianity, and 1.34% were Protestant [15]. The district has 36 rural and 3 urban kebeles [16].

### Source participants

All reproductive age group women were source populations.

### Study population

Sampled women aged between 15–45 years were the study populations.

### Sampling techniques and sample size estimation

A systematic random sampling technique was employed to select households. If more than one eligible woman were found in the households, a lottery method was applied to sample from them. Considering the proportion of cervical cancer screening rate (50%), 95% confidence interval, and 5% margin of error, the required sample size was calculated by using single

population proportion sample size calculation formula.

$$n = \frac{Z^2\,P\,(1-P)}{D^2} = \frac{1.96^2*0.5\,(0.5)}{0.05^2} = 384$$

And considering a 10% non-response rate the total sample size was 422. P-value of 0.5 was taken to obtain optimum sample size, and avoid over-reporting (i.e., SBCC interventions and screening campaigns were conducted at the study setting before the study which might exaggerate the study findings if optimum sample size was not taken). All reproductive age group women who lived at the study setting for greater or equal to six months were considered eligible to participate on the study.

## Data collection procedure

Before preparing the data collection tool, an elicitation study was done by conducting in-depth interviews with 20 participants in the target groups to the locally available salient behavioral, normative, and control beliefs on cervical cancer screening. Then, a structured questionnaire prepared based on TPB constructs to collect data. The questionnaires were administered by trained interviewers.

## Data quality management

Initially, a structured questionnaire was developed in the English language and translated to Afan Oromo and then back to English to maintain its consistencies in meaning and sense by language experts. Then, the questionnaire was checked for validity and reliability conducting a pilot study with 5% of participants at another study area on the population of interest. On top of that, a reliability test was conducted to ensure internal consistency. The Cronbach alpha value of greater than or equal to 0.7 was regarded as an acceptable level. The required corrections in language and content were done for better clarity and more understanding. Furthermore, data quality was assured by conducting intensive supervision, and also checking and assuring quality before electronic data entry. Then, a double data entry method was used in a separate spreadsheet. Statistical control during data analysis was used to reduce the influence of confounding factors.

## Variables

**Dependent variable.** Intention to cervical Cancer Screening
**Independent variable.**

- Socio-demographic characteristics: Age, religion, educational status, occupational status, monthly income, Ethnicity, Marital status.

- TPB constructs: Direct Attitude, direct subjective norm, direct perceived behavioral control, indirect attitude, indirect subjective norm, and indirect perceived behavioral control.

## Measurement

Each of the direct constructs of TPB (direct attitude, direct subjective norm and direct perceived behavioral control) and behavioral intention towards cervical cancer screening was measured using four items with five points of Likert scale. For each constructs, the response variables were calculated by summing up the responses obtained under their four items.

## Statistical analysis

Data were entered in epi data 3.1 and then exported to and analyzed using SPSS 20 statistical software package. Descriptive statistics was done considering the criterion for significance at α

= 0.05. The correlation analysis was done to check the correlation between the direct constructs of TPB and intention; multicollinearity analysis was done to check the correlation between the independent variables [constructs of TPB]. Also, simple and multiple linear analyses were done to identify statistically significant TPB construct with behavioral intention. Further, exploratory principal component analysis (PCA) was done to address the construct validity. The PCA; assumed Varimax rotation with Kaiser normalization for which factor loading less than 0.40 was considered to retain items on their respective factors.

## Ethical approval and consent to participate

Ethical approval was obtained from the Institutional Review Board of Jimma University Institute of health. Official letter of cooperation was taken from the Institute of Health of Jimma University to Jimma Zone Health Department, and then, similarly, the support letter was written from Jimma Zone Health Department to the selected district and from the district to kebeles and the health facilities as well. Written informed consent was obtained from participants after thoroughly explaining the objectives and benefits of a study. Additionally, written consent was also taken from parents for those participants whose age was under 18 years. To ensure confidentiality and anonymity, any personal identifying information on participants was not be collected.

## Results

### Socio-demographic characteristics

A total of 382 women were interviewed with a response rate of 90.5%. The mean age of the respondents was 26.45 years with 4.76 SD. One hundred forty-two (37.2%) were found in the age group of 25–29 years. A large number of respondents 267 (69.9%) were Muslim followed by orthodox 88(23%). One hundred eighty (47.1%) respondents were attended primary education. Most respondents 370 (96.9%) were married. The majority of the respondents 306 (80.1%) were Oromo in their Ethnicity. (Table 1)

Table 1.  Socio-demographic characteristics of participants in Gomma woreda, Jimma zone, Ethiopia, 2019.

| Variable | Category | N (%) | Variable | Category | N (%) |
|---|---|---|---|---|---|
| Age | 15–19 | 25 (6.5) | Occupation | Housewife | 370 (96.9) |
| | 20–24 | 101 (26.4) | | Farmer | 12 (3.1) |
| | 25–29 | 142 (37.2) | | Merchant | 143 (37.4) |
| | 30–34 | 94 (24.6) | | Government employee | 113 (29.6) |
| | ≥35 | 20 (5.2) | | Students | 65 (17.0) |
| Religion | Muslim | 267 (69.9) | | Daily laborer | 46 (12.0) |
| | Orthodox | 88 (23.0) | Ethnicity | Oromo | 306 (80.1) |
| | Protestant | 12 (3.1) | | Amhara | 36 (9.4) |
| | Wakefeta | 10 (2.6) | | Gurage | 29 (7.6) |
| | Catholic | 5 (1.3) | | Yem | 6 (1.6) |
| Educational status | Unable to read & write | 95 (24.9) | | Others* | 5 (1.3) |
| | Able to read & write | 11 (2.9) | Monthly income (ETB) | ≤300 | 71 (18.6) |
| | Primary education | 180 (47.1) | | 301–566 | 53 (13.9) |
| | Secondary and above | 96 (25.1) | | ≥566 | 258 (67.5) |

Note:

*keffa = 3, Tigray = 1, Wolayita = 1

## Direct measure of TPB constructs

Each of the direct constructs of TPB were evaluated by four items and assessed using five point Likert scale measurements. Concerning the frequency of the direct constructs, 177 (46.3%) respondents reported that screening for cervical cancer was good. Similarly, 183 (47.9%) respondents strongly agreed with the usefulness of cervical cancer screening. Around 173 (45.3) and 171 (44.8) of respondents also was agreed that screening for cervical cancer is pleasant and interesting, respectively. The majority of the respondents 309 (80.1%) agreed that most people who are important to them were thought to be screened for cervical cancer. One hundred sixty-eight (44%) respondents reported that they were planned for cervical cancer screening. One hundred forty-three (37.4%) respondents agreed that screening for cervical cancer is under their control. On the other hand, 196 (51.3%) of the respondents reported that it is under their control to screen for cervical cancer. The majority of the respondents, 219 (57.3%), were intended to have screening for cervical cancer within the next three months (Table 2).

## Indirect measure of TPB constructs

The behavioral belief & respective evaluations of each behavioral belief were assessed using five point Likert scale measurements. Based on this measurement, 247 (64.7%) respondents agreed that screening for cervical cancer is important to get better health. The evaluation outcome result showed that 210 (55.0%) respondents responded that screening for cervical cancer is good for getting healthy. Besides these, 190 (49.7%) of the respondents agreed that screening for cervical cancer reduces morbidity among women. On the other hand, the indirect subjective norm was assessed by using five point Likert scale measurements. The result showed that 202 (52.9%) respondents strongly agreed that health extension workers (HEW) think that women should screen for cervical cancer. Three hundred two (79.1%) respondents agreed that their neighbors think pregnant women should screen for cervical cancer. The motivation to

**Table 2. Descriptive statistics for the direct TPB constructs among women in Gomma district, Jimma, Ethiopia (n = 382).**

| Items of direct attitude | Strongly disagree | Disagree | Neutral | Agree | Strongly agree |
|---|---|---|---|---|---|
| Screening for cervical cancer is good. | 1 (0.3) | 9 (2.4) | 22 (5.8) | 177 (46.3) | 173 (45.3) |
| Screening for cervical cancer is useful. | 0 (0) | 10 (2.6) | 22 (5.8) | 167 (43.7) | 183 (47.9) |
| Screening for cervical cancer is pleasant. | 5 (1.3) | 20 (5.2) | 47 (12.3) | 173 (45.3) | 137 (35.9) |
| Screening for cervical cancer is interesting. | 8 (2.1) | 24 (6.3) | 58 (15.2) | 171 (44.8) | 121 (31.7) |
| **Items of direct perceived behavioral control** | **Strongly disagree** | **Disagree** | **Neutral** | **Agree** | **Strongly agree** |
| Screening for Cervical cancer is easy. | 33 (8.6) | 133 (34.8) | 8 (2.1) | 153 (40.1) | 55 (14.4) |
| Screening for Cervical cancer is under my control | 57 (14.9) | 117 (30.6) | 8 (2.1) | 143 (37.4) | 57 (14.9) |
| Screening for Cervical cancer is planned. | 42 (11.0) | 117 (30.6) | 7 (1.8) | 168 (44.0) | 48 (12.6) |
| Screening for Cervical cancer is unconditional | 62 (16.2) | 100 (26.2) | 9 (2.4) | 161 (42.1) | 50 (13.1) |
| **Items of direct subjective norm** | **Strongly disagree** | **Disagree** | **Neutral** | **Agree** | **Strongly agree** |
| Most people approve for me to Screen for Cervical cancer. | 0 (0) | 7 (1.8) | 12 (3. 1) | 287 (75.1) | 76 (19.9) |
| Most people think for me to Screen for Cervical cancer. | 1 (0.3) | 12 (3.1) | 11 (2.9) | 309 (80.9) | 49 (12.8) |
| Most people want to Screen for Cervical cancer. | 1 (0.3) | 14 (3.7) | 8 (2.1) | 291 (76.2) | 68 (17.8) |
| The decision is up to me to have Cervical Cancer Screening. | 11 (2.9) | 116 (30.4) | 17 (4.5) | 196 (51.3) | 42 (11.0) |
| **Items of behavioral intention** | **Strongly disagree** | **Disagree** | **Neutral** | **Agree** | **Strongly agree** |
| I am intended to Screen for cervical cancer within the next 3 months. | 4 (1.0) | 100 (26.2) | 13 (3.4) | 219 (57.3) | 46 (12.0) |
| I will Screened for Cervical cancer in the next 3 months | 9 (2.4) | 93 (24.3) | 21 (5.5) | 211 (55.2) | 48 (12.6) |
| I want to Screen for Cervical cancer in the next 3 months. | 11 (2.9) | 78 (20.4) | 11 (2.9) | 200 (52.4) | 82 (21.5) |
| I like to Screening for Cervical cancer the next 3 months | 15 (3.9) | 65 (17.0) | 5 (1.3) | 200 (52.4) | 97 (25.4) |

comply with assessment also showed that 272 (71.2%) respondents reported that their neighbor's approval for screening for cervical cancer was much important. Also, 208 (54.5%) respondents were reported that their husband's approval was very much important to screening for cervical cancer (Table 3).

**Mean scores of the constructs of TPB.** A descriptive statistical analysis was done to measure the mean score of TPB components. Direct attitude, subjective norm, and perceived behavioral control had a mean score of 16.78 (SD = 2.87), 15.61 (SD = 1.92), and 12.86 (SD = 4.85), respectively. The mean score of intention was 14.52 (SD = 4.012) which approached the maximum value of intention sum score. Indirect attitude, indirect subjective norm, and indirect perceived behavioral control had a mean score of 138.88 (SD = 25.56), 94.96 (SD = 19.10), and 61.71(SD = 21.76), respectively, (Table 4).

## Correlation analysis result

A Pearson's correlation analysis was done to examine the association between components of the TPB model and women intention towards cervical cancer screening. The findings indicated that there was a positive and medium correlation between direct attitude with indirect attitude, and direct perceived behavioral control with indirect perceived behavioral (r = 0.564 & 0.618, p<0.001, respectively). There was also a positive and medium correlation between intention with all attitudes and perceived behavioral control. Intension has a positive and also moderate correlation with direct attitude, subjective norm, and perceived behavioral control. On the other hand, the direct subjective norm has a positive but weak correlation between the direct perceived behavioral controls (Table 5).

## Principal component analysis result

Exploratory principal component analysis (PCA) was done to address the construct of the issue of construct validity. The PCA; assumed Varimax rotation with Kaiser normalization for which factor loading less than 0.40 was considered to retain items on their respective factors. On top; a reliability test was conducted to ensure internal consistency. The Cronbach alpha value of greater than or equal to 0.7 was regarded as an acceptable level (Table 6).

## Regression and collinearity analysis result

Simple linear regression analysis was conducted to assess the association b/n intention and direct constructs of TPB after checking of normality test using a histogram. All the constructs were candidate variables from simple linear regression analysis and entered multiple linear regressions analysis. From multiple linear regression analysis, the direct attitude (B = 0.346, p<0.001), direct subjective norm (B = 0.288, p = 0.008), direct perceived behavioral control (B = 0.132, p = 0.002), indirect attitude (B = 0.015, p = 0.019), indirect subjective norm (B = .017, p = 0.046) and the indirect perceived behavioral control (B = 0.132, p = 0.002) were statistically significant with the intention to screen for cervical cancer. This means a positive unit change in the attitude towards the advantage of cervical cancer screening; intention to the screen was increased by 0.346 units provided that other variables kept constant. For a positive unit change in individuals that approve in screening for cervical cancer, intention to screen for cervical cancer was increased by 0.288 units provided that other variables kept constant. For a positive unit change in perceived controlling of beliefs on environmental/situational facilitators to use cervical cancer screening services, the intention was increased by 0.132 units provided that other variables are kept constant. Similarly, for a positive unit change in an indirect attitude, indirect subjective norm and indirect perceived behavioral control, intention to screen for cervical cancer screening would increase by 0.015, 0.017 and 0.132 units,

**Table 3. Descriptive statistics for the indirect constructs of TPB assessment among women in Gomma district, Jimma, Ethiopia (n = 382).**

| Items of indirect attitude- Behavioral belief | Strongly disagree | Disagree | Neutral | Agree | Strongly agree |
|---|---|---|---|---|---|
| Screening for Cervical cancer is important to get better health. | 0 (0) | 2 (.5) | 4 (1.0) | 247 (64.7) | 129 (33.8) |
| Screening for Cervical cancer helps to get a healthy child. | 0 (0) | 0 (0) | 3 (.8) | 169 (44.2) | 210 (55.0) |
| Screening for Cervical cancer helps to reduce morbidity. | 0 (0) | 6 (1.6) | 12 (3.1) | 190 (49.7) | 174 (45.5) |
| Screening for Cervical cancer helps to reduce the mortality of women. | 10 (2.6) | 100 (26.2) | 139 (36.4) | 90 (23.6) | 43 (11.3) |
| Screening for Cervical cancer helps women to conduct their job freely. | 0 (0) | 17 (4.5) | 22 (5.8) | 230 (60.2) | 112 (29.3) |
| **Items of indirect attitude-evaluation of behavioral belief** | **Very bad** | **Bad** | **Neutral** | **Good** | **Very good** |
| Screening for Cervical cancer is Getting healthy. | 0 (0) | 2 (.5) | 4 (1.0) | 210 (55.0) | 166 (43.5) |
| Getting a healthy child is important. | 0 (0) | 1 (.3) | 1 (.3) | 153 (40.1) | 227 (59.4) |
| Reducing fear of screening is important. | 0 (0) | 6 (1.6) | 7 (1.8) | 219 (57.3) | 123 (32.2) |
| Getting information about Cervical cancer screening important. | 2 (.5) | 46 (12.0) | 16 (4.2) | 195 (51.0) | 123 (32.2) |
| Being free from workload is important for Cervical cancer screening. | 0 (0) | 12 (3.1) | 20 (5.2) | 287 (75.1) | 63 (16.5) |
| **Items of indirect subjective norm-normative belief** | **Strongly disagree** | **Disagree** | **Neutral** | **Agree** | **Strongly agree** |
| My mother thinks I should screen for cervical cancer. | 5 (1.3) | 64 (16.8) | 16 (4.2) | 194 (50.8) | 103 (27.0) |
| My husband thinks that I should screen for Cervical Cancer. | 1 (.3) | 17 (4.5) | 12 (3.1) | 187 (49) | 165 (43.2) |
| HDA leader thinks that I should screen for cervical cancer. | 1 (.3) | 55 (14.4) | 46 (12) | 257 (67.3) | 23 (6.0) |
| My neighbors think that I should screen for cervical cancer. | 1 (.3) | 25 (6.5) | 25 (6.5) | 302 (79.1) | 29 (7.6) |
| Traditional healers think that I should screen for cervical cancer. | 8 (2.1) | 125 (32.7) | 72 (18.8) | 150 (39.3) | 27 (7.1) |
| HEWs think that I should screen for cervical cancer. | 3 (.8) | 5 (1.3) | 4 (1.0) | 168 (44.0) | 202 (522.9) |
| **Items of indirect subjective norm-motivation to comply** | **Not very much** | **Not much** | **Neutral** | **Much** | **Very much** |
| My mother approval my screening for cervical cancer. | 52 (13.6) | 17 (4.5) | 10 (2.6) | 176 (46.1) | 127 (33.2) |
| My husband's approval to screen for cervical cancer. | 4 (1.0) | 6 (1.6) | 13 (3.4) | 151 (39.5) | 208 (54.5) |
| HDA leader approval my screening for cervical cancer. | 1 (.3) | 54 (14.1) | 65 (17.0) | 214 (56.0) | 48 (12.6) |
| My neighbors approval my screening for cervical cancer. | 2 (.5) | 29 (7.6) | 38 (9.9) | 272 (71.2) | 41 (10.7) |
| Traditional healers approval my screening for cervical cancer. | 25 (6.5) | 132 (34.6) | 59 (15.4) | 140 (36.6) | 26 (6.8) |
| HEWs approve my screening for cervical cancer. | 1 (.3) | 4 (1.0) | 6 (1.6) | 185 (48.4) | 186 (48.7) |
| **Items of indirect perceived behavioral control-control belief** | **Strongly disagree** | **Disagree** | **Neutral** | **Agree** | **Strongly agree** |
| To go to the health facility for cervical cancer screening, I have no transportation cost. | 300 (78.5) | 72 (18.8) | 3 (0.8) | 2 (0.5) | 0 (0) |
| When I want to screen, I cannot get consent from my husband. | 341 (89.3) | 33 (8.6) | 4 (1.0) | 3 (0.8) | 1 (.3) |
| To have screening for cervical cancer, I have a workload. | 244 (63.87) | 120 (31.4) | 16 (4.2) | 2 (0.5) | 0 (0) |
| To have screening for cervical cancer, I fear the procedure. | 120 (31.4) | 86 (22.5) | 14 (3.66) | 68 (17.8) | 186 (48.7) |
| To screening for cervical cancer, in our culture, it is forbidden to show our private part to another person other than our husband. | 100 (26.18) | 74 (19.37) | 12 (3.1) | 124 (32.46) | 72 (18.8) |
| When I want to screen, I fear to show my private to another person. | 124 (32.46) | 72 (18.8) | 70 (18.3) | 33 (8.6) | 83 (21.73) |
| **Items of indirect perceived behavioral control- power of control** | **Strongly disagree** | **Disagree** | **Neutral** | **Agree** | **Strongly agree** |
| I have a transportation cost problem to the health facility for screening. | 302 (79.1) | 70 (18.3) | 2 (0.5) | 2 (0.5) | 1 (.3) |
| My husband does not give me consent to screen for cervical cancer. | 324 (84.8) | 52 (13.6) | 2 (0.5) | 2 (0.5) | 2 (0.5) |
| I have the workload to screen for cervical cancer. | 253 (66.23) | 114 (29.84) | 12 (3.1) | 2 (0.5) | 1 (.3) |
| I fear the procedure of cervical cancer screening. | 124 (32.46) | 80 (20.94) | 16 (4.2) | 74 (19.37) | 178 (46.6) |
| Our culture not allows us to show our private part to another person other than our husband. | 100 (26.18) | 74 (19.37) | 12 (3.1) | 124 (32.46) | 72 (18.8) |
| I fear to show my private part during the screening. | 124 (32.46) | 72 (18.8) | 70 (18.3) | 33 (8.6) | 83 (21.73) |

**Table 4. Descriptive statistics for the components of the theory of planned behavior model and women intention in Gomma district, Jimma, Ethiopia (n = 382).**

| Components | No of items | Min. value | Max. value | Mean | SD |
|---|---|---|---|---|---|
| Direct attitude | 4 | 6 | 20 | 16.78 | 2.87 |
| Direct Subjective norm | 4 | 8 | 20 | 15.61 | 1.92 |
| Direct PBC | 4 | 4 | 20 | 12.86 | 4.85 |
| Intention | 4 | 4 | 20 | 14.52 | 4.01 |
| Behavioral belief (BB) | 5 | 15 | 25 | 32.77 | 3.71 |
| Evaluation of behavioral belief (EBB) | 5 | 16 | 25 | 33.09 | 3.48 |
| Indirect attitude = £(BB*EBB) | 5 | 30 | 200 | 138.88 | 25.56 |
| Normative belief(NB) | 6 | 7 | 30 | 23.31 | 2.91 |
| Motivation to comply (MTC) | 6 | 9 | 30 | 23.23 | 3.12 |
| Indirect SN = £(NB*MTC) | 6 | 10 | 150 | 94.96 | 19.10 |
| Control belief (CB) | 6 | 8 | 30 | 18.07 | 4.32 |
| Power of control belief (PCB) | 6 | 8 | 30 | 20.07 | 3.41 |
| Indirect PBC = £(CB*PCB) | 6 | 11 | 136 | 61.71 | 21.76 |

respectively, provided that the other variables are kept constant. On the other hand, from multicollinearity analysis, all variables had no strong correlation in between them (Table 7).

## Discussion

This study found that majority of the respondents 57.3% were intended to have screening for cervical cancer within the next three months from the date of the data collection. Positive direct and indirect attitudes towards cervical cancer screening, positive direct and indirect subjective norm, and direct and indirect perceived behavioral control were factors that affected women's intention to screen for cervical cancer.

The result of this study showed that 57.3% of respondents had the intention to screen for cervical cancer in three months from the date of data collection. This finding is slightly higher than the finding of studies conducted in Ethiopia and Malawi which showed that women's intention for cervical cancer screening was 45.3% and 57.2%, respectively [17]. However, the result is considerably lower than the findings of the study conducted in Uganda and Canada, where 63.0% and 84% of the respondents intended to screen for cervical cancer, respectively [18, 19]. The difference might be due to variation in access to information and screening services. Most of the respondents of the former studies heard about cervical cancer and screening services introduced earlier in those countries.

**Table 5. Pearson's correlation between components of the TPB model and women intention in Gomma district, Jimma, Ethiopia (n = 382).**

| Component | Direct attitude | Indirect attitude | Direct subjective norm | Indirect subjective norm | Direct perceived behavioral control | Indirect perceived behavioral control | Intention |
|---|---|---|---|---|---|---|---|
| Direct attitude | 1 | | | | | | |
| Indirect attitude | .564** | 1 | | | | | |
| Direct subjective norm | .353** | .482** | 1 | | . | | |
| Indirect subjective norm | .297** | .562** | .585** | 1 | | | |
| Direct perceived behavioral control | .311** | .309** | .148** | .172** | 1 | | |
| Indirect perceived behavioral control | .284** | .268** | .153** | .183** | .618** | 1 | |
| Intention | .557** | .502** | .365** | .352** | .492** | .397** | 1 |

"**" the correlation is significant at P < 0.001.

**Table 6. Principal component analysis (PCA) of constructs of theory of planned behavior assuming varimax rotation with Kaiser normalization and factor loading value greater than 0.40.**

| Serial number | Factors | Number of items | Rotated % Variance explained | Factor loading | Crombach alpha | Overall % variance explained |
|---|---|---|---|---|---|---|
| 1 | PBC | 4 | 19.16 | 0.83–0.92 | 0.95 | 79.98% |
| 2 | Intention | 4 | 21.17 | 0.83–0.86 | 0.94 | |
| 3 | Attitude | 4 | 19.32 | 0.78–0.86 | 0.89 | |
| 4 | Subjective norm | 4 | 13.41 | 0.77–0.87 | 0.78 | |

PBC = perceived behavioral control

This study also revealed that a positive attitude towards cervical cancer screening was significantly associated with women's intention towards cervical cancer screening. In this study, 46.3% and 47.9% of respondents reported that using cervical cancer screening is good and very useful, respectively. This study also found that there was a favorable attitude towards using cervical cancer screening because the direct attitude mean score was approaching the maximum mean score. This finding is similar to the finding of studies conducted at different settings which showed that attitude towards cervical cancer screening as the most significant predictor of women's intention towards cervical cancer screening [17, 20]. However, the finding of this study differs from the findings of a study conducted among Latin's which showed that attitude toward cervical cancer screening did not predict the intention to be screened. This might be due to a lack of information on cervical screening, geographical, or population differences. Therefore, this needs an effort that should be exerted to improve the attitude of women towards cervical cancer screening.

This study also found that a positive subjective norm was the other predictors of the women's intention to screen for cervical cancer. In this study, 80.9% of the respondents agreed that most important people [i.e. husbands, parents, HEWs, etc.] approved to them to had cervical cancer screening. This finding is similar with the findings of studies conducted at different settings which indicated that community members, traditional birth attendants, family members, friends, husbands, and health care providers play an important role in creating women's awareness and deciding to screen for cervical cancer [17, 20, 21]. Thus, this calls a need to take action to improve or change the attitude and behavior of influential people within the community to improve women's intention for cervical cancer screening.

Direct perceived behavioral control was also another predictor of intention. Similarly, the finding of studies conducted at different settings indicated that women's perceived behavioral control was one predictor for having cervical cancer screening [17, 20, 21]. In this study, 31.9%

**Table 7. Independent factors associated with behavioral intention to cervical cancer screening, Gomma district, Jimma, Ethiopia, 2019 (n = 382).**

| Model | Unstandardized Coefficients | | Standardized Coefficients | t | Sig. | Multicollinearity statics | |
|---|---|---|---|---|---|---|---|
| | B | Std. Error | Beta | | | Tolerance | VIF |
| (Constant) | -4.637 | 2.384 | | -1.945 | .053 | 0.88 | 5.67 |
| Direct attitude | .346 | .077 | .244 | 4.523 | .000** | 0.56 | 4.97 |
| Indirect attitude | .015 | .009 | .095 | 1.703 | .019** | 0.33 | 3.29 |
| Direct SN | .288 | .107 | .138 | 2.686 | .008** | 0.77 | 4.99 |
| Indirect SN | .017 | .012 | .080 | 1.390 | .046** | 0.17 | 4.40 |
| Direct PBC | .132 | .043 | .168 | 3.068 | .002** | 0.47 | 2.17 |
| Indirect PBC | .030 | .009 | .166 | 3.189 | .002** | 0.19 | 4.89 |

** Statistically significant predictors at P<0.05. VIF: variance inflation factor

and 27.7% of respondents were reported that cervical cancer screening was difficult and not under their control, respectively. This might be due to women who thought they may not get individuals that can give care for their family when they went for screening. This calls a need for interventions to increase women's sense of control to undergo or improve their cervical cancer screening.

## Conclusions

From this study it was understood that women's intention towards cervical cancer screening was low. The predictors of women's behavioral intention to screen for cervical cancer were the direct and indirect attitude, direct and indirect subjective norm, direct and indirect perceived behavioral control. Thus, this calls a need to develop strategies and take action to improve the attitude of women and influential peoples around them towards cervical cancer screening and increase their sense of control to improve women's intention for cervical cancer screening. Moreover, health care providers should have to conduct behavioral change communication focusing on the constructs of the theory of planned behavior.

### Strength and limitation of the study

The study applied well the concept of the theory of planned behavior in determining the women's intention and the predictors of behavioral intention. As a limitation, the study could not consider other variables like socio-demographics, knowledge about cervical cancer, or previous history of cervical cancer screening as predictors of the behavioral intention. There might be a possibility for bias to emerge in systematic sampling, since the samples were selected by randomly throwing into the air and the researcher uses his own discretion.

## Acknowledgments

We acknowledge the study participants for their voluntary participation.

## Author Contributions

**Conceptualization:** Wadu Wollancho, Demuma Amdissa, Shemsedin Bamboro, Yitbarek Wasihun, Kasahun Girma Tareke, Abraham Tamirat Gizaw.

**Data curation:** Wadu Wollancho, Demuma Amdissa, Shemsedin Bamboro, Yitbarek Wasihun, Kasahun Girma Tareke, Abraham Tamirat Gizaw.

**Formal analysis:** Wadu Wollancho, Demuma Amdissa, Shemsedin Bamboro, Yitbarek Wasihun, Kasahun Girma Tareke, Abraham Tamirat Gizaw.

**Funding acquisition:** Wadu Wollancho, Demuma Amdissa, Shemsedin Bamboro, Yitbarek Wasihun, Kasahun Girma Tareke, Abraham Tamirat Gizaw.

**Investigation:** Wadu Wollancho, Demuma Amdissa, Shemsedin Bamboro, Yitbarek Wasihun, Kasahun Girma Tareke, Abraham Tamirat Gizaw.

**Methodology:** Wadu Wollancho, Demuma Amdissa, Shemsedin Bamboro, Yitbarek Wasihun, Kasahun Girma Tareke, Abraham Tamirat Gizaw.

**Project administration:** Wadu Wollancho, Demuma Amdissa, Shemsedin Bamboro, Yitbarek Wasihun, Kasahun Girma Tareke, Abraham Tamirat Gizaw.

**Resources:** Wadu Wollancho, Demuma Amdissa, Shemsedin Bamboro, Yitbarek Wasihun, Kasahun Girma Tareke, Abraham Tamirat Gizaw.

**Software:** Wadu Wollancho, Demuma Amdissa, Shemsedin Bamboro, Yitbarek Wasihun, Kasahun Girma Tareke, Abraham Tamirat Gizaw.

**Supervision:** Wadu Wollancho, Demuma Amdissa, Shemsedin Bamboro, Yitbarek Wasihun, Kasahun Girma Tareke, Abraham Tamirat Gizaw.

**Validation:** Wadu Wollancho, Demuma Amdissa, Shemsedin Bamboro, Yitbarek Wasihun, Kasahun Girma Tareke, Abraham Tamirat Gizaw.

**Visualization:** Wadu Wollancho, Demuma Amdissa, Shemsedin Bamboro, Yitbarek Wasihun, Kasahun Girma Tareke, Abraham Tamirat Gizaw.

**Writing – original draft:** Kasahun Girma Tareke.

**Writing – review & editing:** Wadu Wollancho, Demuma Amdissa, Shemsedin Bamboro, Yitbarek Wasihun, Kasahun Girma Tareke, Abraham Tamirat Gizaw.

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
