## [Decision Letter · Decision Letter 0]

2 Aug 2020

PONE-D-20-14038

Determining behavioral intention and its predictors towards cervical cancer screening among women in Gomma district, Jimma, Ethiopia: Application of the theory of planned behavior

PLOS ONE

Dear Dr. Tareke,

Thank you for submitting your manuscript to PLOS ONE. After careful consideration, we feel that it has merit but does not fully meet PLOS ONE’s publication criteria as it currently stands. Therefore, we invite you to submit a revised version of the manuscript that addresses the points raised during the review process.

We look forward to receiving your revised manuscript.

Kind regards,

Amir H. Pakpour, Ph.D.

Academic Editor

PLOS ONE

Journal Requirements:

3. We note you have included a table to which you do not refer in the text of your manuscript. Please ensure that you refer to Table 7 in your text; if accepted, production will need this reference to link the reader to the Table.

Reviewers' comments:

Reviewer's Responses to Questions

**Comments to the Author**

1. Is the manuscript technically sound, and do the data support the conclusions?

Reviewer #1: Yes

2. Has the statistical analysis been performed appropriately and rigorously? 

Reviewer #1: Yes

3. Have the authors made all data underlying the findings in their manuscript fully available?

Reviewer #1: Yes

4. Is the manuscript presented in an intelligible fashion and written in standard English?

Reviewer #1: Yes

5. Review Comments to the Author

Reviewer #1: Dear author(s),

I appreciate you due to all the efforts. This manuscript is overall reasonable and admirable, short of a few points recommended below:

1.Please apply MeSH part on pubmed.com and extract the most related keywords.

2.Why was systematic sampling chosen?There is a possibility for bias to emerge in systematic sampling, if the researcher throws the randomness into air and uses his own discretion in selection of items in framing the sample.(write this in limitation of your study)

3. Your background is a bit long. Please transmit a part of them to discussion part.(if applicable)

4.Why were not considered eligible criteria (inclusion and exclusion criteria). Some of studies indicated that cervical cancer highly likely depend on marital status and sexual relation and age. Is there any corroboration to exist cervical cancer in women aged 15?

5.why did you consider prevalence(p) criterion in your formula 0.5? is there any evidence for it?please report.

6. In data management tools, please report the criterion to confirm reliability and validity.

7. why did you mention to PCA in Data quality management?explain or omit this part.

8.In abstract part, you mentioned to multiple linear regression model and in statistical analysis gave logistic regression. why? As well, in your Abstract-Results, use either plus-minus or SD in parentheses.

9.Please bring Cronbach's alpha and Ethical approval in method part. they are not related to statistical analysis.

10.How did you calculate Intention to cervical Cancer Screening response variable. please express the type of each variable. (predictors-response)

11.presentation of descriptive analysis for each predictor and its items is dispensable. please reduce some of additional tables.

12.please check the presumptions of using Pearson correlation and report in your manuscript.

Although I wrote this points, however, I really appreciate and admire you due to their interpretation of statistical analysis. It was really great. Thank you so much.

With best wishes,

Maryam Ganji

6. PLOS authors have the option to publish the peer review history of their article (what does this mean?). If published, this will include your full peer review and any attached files.

Reviewer #1: **Yes: **Maryam Ganji

---

## [Author Response · Author response to Decision Letter 0]

8 Aug 2020

Dear PLOS ONE academic editor and reviewer,

We want to express our deepest gratitude for reviewing and providing your constructive comments to us on a manuscript entitled “Determining behavioral intention and its predictors towards cervical cancer screening among women in Gomma district, Jimma, Ethiopia: Application of the theory of planned behavior” submitted to PLOS ONE journal. We have gone through the reviewer comments and made corrections or amendments accordingly. Please find below our responses based on each comments. 

1. Please apply MeSH part on pubmed.com and extract the most related keywords.

Amendments were done based on the comment. Abstract, keywords, Page 2. 

2. Why was systematic sampling chosen? There is a possibility for bias to emerge in systematic sampling, if the researcher throws the randomness into air and uses his own discretion in selection of items in framing the sample.(write this in limitation of your study)

A systematic sampling technique was used to select the samples because there was no sampling frame containing all eligible reproductive age group women in the study setting. 

Amendment was done based on the reviewer comment writing the suggestion under strength and limitation of the study. Page 16 

3. Your background is a bit long. Please transmit a part of them to discussion part.(if applicable)

Yes, it is true that the background is long. However, we wrote exhaustively to make it informative enough for the readers, and to show clearly the burden of the disease, suggested interventions, activities on the implementation, the research gaps observed from existing evidences and what was going to be done. Therefore, while all these things were incorporated it becomes a little bit long. But, as per the reviewer suggestion, some amendments were one on paragraph 3 and 4. 

4. Why were not considered eligible criteria (inclusion and exclusion criteria). Some of studies indicated that cervical cancer highly likely depends on marital status and sexual relation and age. Is there any corroboration to exist cervical cancer in women aged 15?

It is true that existing evidence revealed that cervical cancer is highly likely depends on marital status and sexual relation and age. However, there are also evidenced that indicated premature sex is a risk for cervical cancer, and the WHO recommends all reproductive age group (15-45 years old) to be screened for cervical cancer. On the other hand, in our country, these days, most of school students at this age start sexual relationship, even with multiple partners. Therefore, to draw scientific conclusions, it is a good idea to include, rather than including them.

Second, we have added the eligibility criteria on page 5 along with sampling technique and sample size calculation. 

5. Why did you consider prevalence (p) criterion in your formula 0.5? Is there any evidence for it? Please report.

It is true that there were evidences indicating the prevalence of cervical cancer screening. However, the prevalence was low to obtain adequate sample size. On the other hand, SBCC interventions and screening campaigns were conducted at the study setting, as indicated on the background part. Therefore, to obtain optimum sample size and avoid over-reporting that might happen due to these cases, p-value of 0.5 was taken. We have added the explanation on page 5. 

6. In data management tools, please report the criterion to confirm reliability and validity.

A reliability test was done and a Cronbach alpha value of greater than or equal to 0.7 was regarded as an acceptable level. Page 6 

7. Why did you mention to PCA in Data quality management? Explain or omit this part.

We omitted. 

8. In abstract part, you mentioned to multiple linear regression model and in statistical analysis gave logistic regression. Why? As well, in your Abstract-Results, use either plus-minus or SD in parentheses.

It was a typographical error, and correction was done on statistical analysis, and changed into linear regression. Page 6.

9. Please bring Cronbach's alpha and Ethical approval in method part. They are not related to statistical analysis. 

We have moved cronbach’s alpha to data quality management. But, the ethical approval part is putted as a method section or part next to the statistical analysis. It is not part of the statistical analysis. Therefore, considering the PLOS ONE manuscript submission format, it is written at the end of the method part. 

10. How did you calculate Intention to cervical Cancer Screening response variable? Please express the type of each variable. (predictors-response) 

It was measured using four items with five points of Likert scale. The four items were summed up and used for analysis. Again, all the direct constructs (variables) were measured using four items with five points of Likert scale; the four items were summed up and used for the analysis. Therefore, we calculated intension for each response variable by summing up the responses obtained under them. 

11. Presentation of descriptive analysis for each predictor and its items is dispensable. Please reduce some of additional tables.

It is true that a lot of tables were presented in the manuscript. We tried to reduce it, but we felt that if one table would be removed, it becomes less informative for the readers, especially for such theory-based researches. Therefore, we kept as it was. 

12. Please check the presumptions of using Pearson correlation and report in your manuscript.

A Pearson correlation analysis is done to understand the association between two continuous variables. All of the variables mentioned under theory of planned behavior and reported in this manuscript were considered as continuous variables. Therefore, a Pearson correlation was done to assess the association between constructs of theory of planned behavior used to determine the predictors of intention towards cervical cancer screening. Page 12. 

Saying this, I hope that the comments provided by the reviewer were addressed and the manuscript would meet the high standards of your journal. Therefore, am looking forward to receive a favorable response from you regarding the acceptance of the manuscript.

Sincerely yours

Kasahun Girma Tareke (corresponding author)

Address: Department of Health, Behavior and Society, Faculty of Public Health, Institute of Health, Jimma University, Jimma, Ethiopia

E-mail: kasahungirmadera@gmail.com; girma.tareke@ju,edu.et; 

Phone: +251 919375374

---

## [Editor Report · Decision Letter 1]

18 Aug 2020

Determining behavioral intention and its predictors towards cervical cancer screening among women in Gomma district, Jimma, Ethiopia: Application of the theory of planned behavior

PONE-D-20-14038R1

Dear Dr. Tareke,

We’re pleased to inform you that your manuscript has been judged scientifically suitable for publication and will be formally accepted for publication once it meets all outstanding technical requirements.

Kind regards,

Amir H. Pakpour, Ph.D.

Academic Editor

PLOS ONE
---

## [Editor Report · Acceptance letter]

19 Aug 2020

PONE-D-20-14038R1 

Determining behavioral intention and its predictors towards cervical cancer screening among women in Gomma district, Jimma, Ethiopia: Application of the theory of planned behavior 

Dear Dr. Tareke:

I'm pleased to inform you that your manuscript has been deemed suitable for publication in PLOS ONE. Congratulations! Your manuscript is now with our production department. 

Kind regards, 

on behalf of

Dr. Amir H. Pakpour 

Academic Editor

PLOS ONE